# Enhancing Liquefaction Efficiency: Exploring the Impact of Pre-Hydrolysis on Hazelnut Shell (*Corylus avellana* L.)

**DOI:** 10.3390/ma17112667

**Published:** 2024-06-01

**Authors:** Luísa Cruz-Lopes, Joana Duarte, Yuliya Dulyanska, Raquel P. F. Guiné, Bruno Esteves

**Affiliations:** 1CERNAS (Centre for Natural Resources, Environment and Society), Polytechnic University of Viseu, Av. Cor. José Maria Vale de Andrade, 3504-510 Viseu, Portugal; lvalente@estgv.ipv.pt (L.C.-L.); ydulyanska@esav.ipv.pt (Y.D.); raquelguine@esav.ipv.pt (R.P.F.G.); 2Department of Environmental Engineering, Polytechnic University of Viseu, Av. Cor. José Maria Vale de Andrade, 3504-510 Viseu, Portugal; joanaduarte62@hotmail.com; 3Department of Food Industry, Polytechnic University of Viseu, Av. Cor. José Maria Vale de Andrade, 3504-510 Viseu, Portugal; 4Department of Wood Engineering, Polytechnic University of Viseu, Av. Cor. José Maria Vale de Andrade, 3504-510 Viseu, Portugal

**Keywords:** hazelnut shells, liquefactions, *Corylus avellana* L., ecovalorization, waste, FTIR-ATR

## Abstract

Hazelnut shells (HS), scientifically known as *Corylus avellana* L. shells, are waste produced by companies that process nuts. The main objective of this study was to find an efficient way to maximize the chemical potential of HS by solubilizing the hemicelluloses, which could then be used to recover sugars and, at the same time, increase the lignin content of this material to produce adhesives or high-strength foams. In order to optimize the pre-hydrolysis process, two different temperatures (160 and 170 °C) and times varying from 15 to 180 min were tested. All the remaining solid materials were then liquefied using polyalcohols with acid catalysis. The chemical composition of hazelnut shells was determined before and after the pre-hydrolysis. All of the process was monitored using Fourier Transform Infrared Spectroscopy with Attenuated Total Reflectance (FTIR-ATR) by determining the spectra of solids and liquids after the pre-hydrolysis and liquefaction steps. The highest solubilization of hazelnut shells was found for 170 °C and 180 min, resulting in a 25.8% solubilization. Chemical analysis after the hydrolysis process showed a gradual increase in the solubilization of hemicelluloses as both the temperature and time of the reactor were increased. Simultaneously, the percentages of α-cellulose and lignin in the material also increased with rises in temperature and duration. FTIR-ATR allowed for the detection of significant spectral changes in the hazelnut shells from their initial state to the solid residue and further into the liquefied phase. This confirmed that pre-hydrolysis was effective in enhancing the chemical composition of the material, making it more suitable for the production of adhesives, polyurethane foams, or in the production of bioplastics and composite materials, combined with other biopolymers or synthetic polymers to enhance the mechanical properties and biodegradability of the resulting materials.

## 1. Introduction

Hazelnuts are widely used in food products due to their flavor, texture, and high nutritional value, including proteins, monounsaturated fatty acids, fibers, vitamins, and bioactive compounds [1]. Hazelnuts are recommended in moderate doses as a healthy nut because their components, such as monounsaturated fatty acids, fibers, and antioxidants, contribute to the prevention of cardiovascular diseases [2], reduction in cholesterol [3], improvement in the lipid profile, and combating of oxidative stress, as well as having anti-inflammatory, neuroprotective, and anticancer properties [4]. In Portugal, hazelnut production is vital for the rural economy, and has been the subject of research to improve production and processing in the sector [5].

In 2022, in Portugal, 5.6 thousand tons were produced [6]. Globally, in the past decades, the area of hazelnut cultivation has been increasing worldwide, with Turkey, the United States of America, Azerbaijan, and Chile being the top producers of this nut. Considering the evolution over time of hazelnut production, it is observed that in 2022, a maximum world production of 1220 thousand tons was achieved, corresponding to a cultivation area of 1073 thousand ha.

Hazelnut byproducts include the shells that are discarded as residues, but which can have alternative utilizations providing added-value products. The production of hazelnut shells was, in 2022, 330 thousand tons, as shown in Figure 1, gathering data from the Portuguese Institute of Statistics [6]. The most relevant regions for the hazelnut shell distribution include Turkey, Kyrgyzstan, Azerbaijan, Chile, Italy, Spain, China, Portugal, France, and certain parts of the USA [6], as depicted in Figure 2.

Hazelnut shells are composed of structural and non-structural components, similar to most lignocellulosic materials. Non-structural components are composed of extractives, which are small organic molecules and ashes that correspond to the inorganic materials that remain after the burning of the material. Structural compounds are cellulose, hemicelluloses, and lignin. Knowing the structure and properties of this material is essential for several applications in many fields, including biomaterials, biofuels, and sustainable materials.

Pre-hydrolysis plays a vital role in biomass processing, serving as preparation for subsequent enzymatic or chemical hydrolysis essential for biofuel or biochemical production. Pre-hydrolysis methods can be conducted with acid or alkaline catalysts or just with water, also called auto-hydrolysis [7]. Acid pre-hydrolysis, typically employing sulfuric acid as a catalyst, results in the production of oligomeric and monomeric sugars; nevertheless, this method has been described to corrode equipment and induce extensive lignin condensation [7]. Additionally, there is a significant cellulose hydrolysis, which is not desirable [8]. Alkaline pre-hydrolysis is commonly conducted using green or white liquor from the kraft process, employing strong alkaline solutions at low temperatures [9]. Water pre-hydrolysis, commonly referred to as auto-hydrolysis, is the preferred pretreatment method where biomass is exposed to compressed hot water, facilitating significant solubilization of hemicelluloses [10]. The primary reaction during this process is the depolymerization of hemicelluloses, which results in the production of sugars and oligosaccharides [11]. This reaction is driven by hydronium ions formed from the autoionization of water [12] and the organic acids produced in situ from the cleavage of acetyl groups. At the same time, these acids help break down the glycosidic bonds in hemicelluloses, leading to their conversion into lower molecular weight polysaccharides, oligosaccharides, monosaccharides, and byproducts such as furfural and hydroxymethylfurfural. Moreover, pre-hydrolysis enhances the surface area and reduces the crystallinity of cellulose, making it more amenable to subsequent hydrolysis processes. The amount of hemicelluloses removed increases over time, but at the same time, some of the xylose produced is converted into furfural. Therefore, the time and temperature of pre-hydrolysis need to be carefully optimized to maximize sugar recovery while minimizing the conversion to furfural [13]. If high temperatures are used, the lignin structure can break and release phenolic compounds that are dissolved in the autohydrolysis liquors [14].

Polyalcohol liquefaction has been used for many years for the production of polyols that can later be converted into value-added materials. Liquefaction can be conducted using basic or acid catalysts. Basic catalysts have been used essentially with cork-based barks due to the high amount of suberin in these materials, since a higher temperature is needed to liquefy the remaining lignocellulosic materials with a basic catalyst [15]. Therefore, a basic catalyst is only recommended for cork oak [16] or cork-rich barks such as Douglas fir bark [17]. Using acid-catalyzed liquefaction, lignocellulosic materials can be almost totally converted [18]. Several works have reported acid-catalyzed liquefaction of different lignocellulosic materials, like, for example, agricultural wastes such as wheat straw [19] or cornstarch [20], forest management residues [21], barks [22], or shells [23] to be later used mainly for the production of polyurethane foams [24] and adhesives [25].

A recent study on acid-catalyzed lignin liquefaction for the production of polyols and polyurethane foams showed that the acid catalysts have positive effects on the polymerization of the PU foam and yields a more hydrophilic and rigid cell structure [24].

This work studied two different pre-hydrolysis temperatures to enhance the potential of hazelnut shells as a sustainable material for chemical production by increasing the lignin content of the material and, at the same time, recovering hemicelluloses. This will allow for making the most of hazelnut shells that, after liquefaction by polyalcohols, such as glycerol and ethylene glycol that can also be obtained from lignocellulosic materials, can be used for the production of foams or adhesives with improved properties.

## 2. Materials and Methods

### 2.1. Materials

The HS used in this work were obtained from Transagri, Lda., a company located in the industrial zone of Mangualde (Mangualde, Portugal). Previous to analysis, the samples were milled using a Retsch SMI mill (Retsch GmbH, Haan, Germany) followed by sieving with a Retsch AS200 (Retsch GmbH, Haan, Germany) for 20 min at 50 rpm. Four fractions were obtained: >40 mesh, 40–60 mesh, 60–80 mesh, and <80 mesh. The 40–60 mesh fraction was the one used for hydrolysis and chemical analysis.

### 2.2. Pre-Hydrolysis

Experiments were carried out using a PARR LKT PED cylindrical glass reactor(Parr Instrument Company, Moline, USA) that holds 600 mL and is double-coated for better thermal insulation. A 20 g sample (40–60 mesh) was mixed with 200 mL of distilled water in the reactor. An automatic stirrer was set to 70 rpm to achieve a uniform mixture. The process involved fine-tuning the temperature (between 160 and 170 °C, regulated by the oil temperature in the reactor’s jacket) and the duration (ranging from 30 to 180 min) to find the optimal conditions for maximum solubilization. It took approximately 15 min for the temperature inside the reactor to achieve the desired temperature. The pressure inside the reactor was around 4 bar. Following the reaction, the contents were removed from the reactor and filtered through a Buckner funnel with a paper filter to separate the solid residue from the liquid fraction. The solid remains were then oven-dried and weighed to calculate the percentage of material that had been solubilized, as shown in Equation (1).
(1)$$Solubilization\left(\%\right)=\frac{Initial\;dry\;mass\left(g\right)-Solid\;dry\;residue(g)}{Initial\;dry\;mass\left(g\right)}\times 100$$

The chemical composition of the solid fraction was analyzed using dried samples from the pre-hydrolysis process. This analysis involved determining the levels of Klason lignin, holocellulose, α-cellulose, and hemicelluloses in the 40–60 mesh fraction. It was presumed that most extractives had been removed during the pre-hydrolysis stage.

### 2.3. Liquefaction

Similarly to pre-hydrolysis, the liquefaction was conducted in a PARR LKT PED cylindrical glass reactor (Parr Instrument Company, Moline, USA) with a 600 mL capacity. A 10 g pre-hydrolyzed sample was combined with 200 mL of distilled water within the reactor. The automatic stirrer was then adjusted to 70 rpm to ensure homogeneity. The temperature used was 180 °C for 60 min. After reactor extraction, the resultant mixture was filtered and weighed similarly to pre-hydrolysis, as shown in Equation (2).
(2)$$Liquefaction\;Yield\left(\%\right)=\frac{Initial\;dry\;mass\left(g\right)-Solid\;dry\;residue(g)}{Initial\;dry\;mass\left(g\right)}\times 100$$

### 2.4. Chemical Composition

The chemical characterization of hazelnut shells was carried out using standard procedures to measure the contents of ash, extractives (via dichloromethane, ethanol, and hot water), lignin, holocellulose, α-cellulose, and hemicellulose. This analysis was performed to evaluate their potential applications.

The 40–60 mesh fraction was dried at 105 °C for a minimum of 24 h before chemical analyses, adhering to the procedures delineated in Tappi T 264 om-97 [26]. The chemical composition analysis of each sample was conducted in triplicate to ensure the strength and reliability of the results.

The ash content of the hazelnut shell was determined using the Tappi T 211 om-93 standard procedure [27], involving the calcination of the material at 525 °C. This method allows for the quantification of the inorganic constituents present in the biomass. To analyze the inorganic composition, the ash obtained from calcination was subjected to wet digestion in a Leco CHNS-932 Elemental Analyzer (St. Joseph, MI, USA), followed by analysis using Inductively Coupled Plasma (ICP) spectroscopy. This approach provides insights into the elemental composition of the ash fraction.

Extractives were quantified via sequential extraction with solvents of ascending polarity (Dichloromethane, ethanol, and water), enabling a thorough evaluation of their distribution within the hazelnut shell matrix. The extraction was made through Soxhlet extraction in accordance with Tappi T 204 om-88 [28]. Approximately 10 g of the dried sample was subjected to Soxhlet extraction, employing 150 mL of each solvent. Dichloromethane extraction was performed for 6 h, while ethanol and water extractions lasted 16 h, following a sequential extraction with increasing polarity. The extractive content was quantified relative to the dry material.

The lignin content in HS free of extractives was quantified using the Klason method, using 72% H_2_SO_4_ as per Tappi T 204 om-88 standards [28]. The procedure consisted of two sequential hydrolysis steps: initially using 72% H_2_SO_4_ for 1 h, followed by a second hydrolysis with 3% H_2_SO_4_ in an autoclave at 120 °C for another hour. The insoluble residue that resulted was gathered through filtration using a G2 glass crucible and dried until it reached a constant weight. Following this, the soluble lignin content was quantified spectrophotometrically by measuring its absorbance at 205 nm.

The holocellulose content was determined using the acid chloride method. This process involved treating the sample in a water bath with a mixture of sodium chlorite solution prepared by dissolving 8.5 g in distilled water and diluting to a final volume of 250 mL and 13.5 g of NaOH dissolved in 50 mL of distilled water, to which 37.5 mL of glacial acetic acid were added, and the final volume was adjusted to 250 mL, at 70 °C until it turned white. The sample was then filtered and dried. The insoluble material that remained was also filtered and subsequently dried [29].

The α-cellulose content was determined by hydrolyzing holocellulose with 2.5 mL of 17.5% NaOH in a thermal bath. After hydrolysis, the insoluble residue was filtered out and then thoroughly washed with 8.3% NaOH and distilled water. A final treatment involved using 3.75 mL of 10% CH_3_COOH in a G2 glass crucible. The residue was then dried to a constant weight. The hemicellulose content was subsequently calculated as the difference between the original holocellulose and the α-cellulose.

### 2.5. FTIR Analysis

Fourier transform infrared spectroscopy (FTIR) was used to analyze the functional groups present in untreated hazelnut shell samples, as well as those subjected to pre-hydrolysis, the liquefied material, and the solid residue post-liquefaction. The original dried hazelnut shells, the liquefied fraction, and the remaining solid residue were prepared for FTIR analysis using the attenuated total reflection (ATR) technique. To ensure the complete removal of moisture, the samples were crushed and dried in an oven at 100 °C for one week prior to analysis. The FTIR-ATR spectra were recorded using a Perkin Elmer UATR Spectrum Two instrument (Waltham, MA, USA). The instrument operated at a scan rate of 72 scans per minute and a resolution of 4.0 cm^−1^, covering a spectral range from 4000 to 400 cm^−1^. After correcting for background, each sample was placed onto the ATR crystal for analysis. Solid samples were pressed firmly against the crystal surface to ensure optimal contact. The spectral data collected were an average of three individual measurements per sample.

## 3. Results and Discussion

### 3.1. Chemical Composition

Table 1 presents the results of the chemical composition analysis of HS. The findings indicate that ashes constitute around 3.6% of the composition, while total extractives make up about 3.6%. The majority of extractives are soluble in water (3.0%) and ethanol (0.4%), with a smaller portion soluble in dichloromethane (0.3%). The total lignin content is determined to be 37.6%, predominantly consisting of insoluble lignin (36.7%), with a minor soluble fraction (0.9%). α-Cellulose content is approximately 33.7%, while hemicelluloses contribute 29.6% to the composition. This comprehensive analysis of chemical constituents offers valuable insights into potential applications of hazelnut shells, guiding future utilization strategies. The chemical composition of hazelnut shells seems to differ between provenances. Turkish hazelnut shells’ chemical composition was presented by Demirbas [30], who obtained 43% lignin, 26% cellulose, and 30% hemicelluloses. Similarly, Rivas et al. [31] reported 40% lignin, 26% Glucose (Glucan), and approximately 32% hemicelluloses for hazelnut shells from Ourense, Spain. Our samples presented a smaller amount of lignin of around 37.6% (38.2% excluding the extractives), a higher amount of cellulose, 33.7%, and a lower amount of hemicelluloses (29.6%). Hazelnut shells from Italy (Viterbo, Rome) presented a similar amount of lignin (36.6%), a lower amount of hemicelluloses (23.6%), and cellulose (22.9%), although these authors accounted for one more fraction representing 8% which they called “proteinaceous fraction” [32]. Hoşgün and Bozan [33] presented a very different chemical composition of hazelnut from turkey with 51.3% lignin, 16.7% cellulose, 13.3% hemicelluloses, 5.1% extractives, 2.1% ash, and 11% of compounds called others which they did not mention which kind of compounds were and how they were obtained. Although differences between regions and even in the same region are expected due to natural variability, some of the differences observed can be due to the different methods used for the chemical characterization of the material. The main differences are observed in hemicelluloses’ content, since, in most cases, they are determined as a difference between holocellulose and cellulose content. Also the methods for cellulose determinations differ a lot between works. For example Hoşgün and Bozan [33] and Rivas et al. [31] use the monosaccharides composition to estimate the amount of cellulose and hemicelluloses, while Demirbas [30] uses the difference between hollocellulose and alfa-cellulose.

Information on the inorganics present in ash is important in several fields, such as when hazelnut shells are used for energy production, since the ash content affects the efficiency and cleanliness of these processes. High ash content can lead to slagging, fouling, and equipment corrosion, reducing efficiency and increasing maintenance costs. High concentrations of Ca, Na, and K have been reported to lower the deformation temperatures of materials, which, according to the ENPlus^®^ standard, must exceed 1200 °C for A1 category wood pellets. Nevertheless, the levels of these elements are significantly lower than those reported before for several shrub species [34]. More importantly, in order to valorize ash, depending on its composition, ash can have potential applications in agriculture, construction materials, or as a source of valuable minerals. The main inorganic compounds found in HS ash were potassium, with 27.7%, followed by calcium 16.9%, magnesium 1.4%, and sodium 0.3%. Ash composition is also very erratic, since other authors reported different chemical compositions. For instance, Baran et al. [35] stated that the main compounds of an HS originating from Turkey were calcium, followed by potassium, sodium, and silicon, while Yurt and Bekar [36] obtained higher amounts of calcium, potassium, and silicon followed by iron and magnesium. HS from Serbia had mainly K (26.29%), Ca (11.62%), Mg (6.7%), and P (6.10%) [37].

The pre-hydrolysis using water as the solvent was performed before liquefaction in order to remove part of the hemicelluloses that can later be converted into usable oligosaccharides. For example, Surek and Buyukkileci [38] tested several hazelnut residues and concluded that the shells exhibited the highest xylan content at 18.7%, compared to other byproducts of hazelnut processing. As a result, they yielded a greater production of xylooligosaccharides (XOS). With a pre-hydrolysis at 190 °C for 15 min, Surek et al. [39] obtained a maximum low-DP-XOS of 22.5 g/L. The optimal temperature for autohydrolysis to produce soluble hemicellulosic oligosaccharides was determined to be 210 °C in accordance with Rivas et al. [31]. According to Hoşgün and Bozan [33], who studied different pre-hydrolysis methods for hazelnut shells, the maximum hemicellulose removal was around 15% at 120 °C, 22% at 150 °C, and 90% at 200 °C. Although most of the hemicelluloses were removed at 200 °C, a significant amount of cellulose of approximately 15–20% and about 18% of lignin were also removed. Alkaline pretreatment was the most damaging to lignin, removing almost 80%.

The pre-hydrolysis was made at two different temperatures, 160 °C and 170 °C, and at six times (ranging from 15 to 180 min). The liquefaction yield is presented in Figure 3 as a function of time for each temperature studied. Liquefaction yield increased both with time and temperature. For instance, at 160 °C, the liquefaction yield was 3.9% after 30 min and 17.5% after 180 min, while at 170 °C, the liquefaction yield was 25.8% for 180 min, higher than the yield obtained at 160 °C. Several authors reported the autohydrolysis of hazelnut shells at different temperatures, 190 °C (15 min) [39] or 210 °C [31], but did not mention the solubilization percentage. Similar temperatures were used by Surek et al. [29] for walnut shells and presented higher percentages of solubility at 170 °C with around 23% (15 min) and 29% (30 min). These percentages of solubility could only be obtained for around 150 min and 180 min for hazelnut shells. These results prove that higher temperatures result in higher amounts of hemicelluloses removal, and the results from the chemical composition of hydrolyzed material show that up to 170 °C and 180 min hydrolysis (Figure 4), there is no significant attack on lignin since percentage increased for higher temperature and time of hydrolysis. The higher hydrolyzation time seems, however, to affect alfa-cellulose content since it decreases for 180 min for both temperatures studied.

The objective of pre-hydrolysis is the removal of sugars without significantly affecting lignin and cellulose. Figure 4 presents the chemical composition variation over the several pre-hydrolysis steps at both temperatures of 160 °C and 170 °C.

An increase in lignin content relative to the initial sample is observed along the pre-hydrolysis for both temperatures. At 160 °C, the initial 36.7% content increases to around 48.3% after 180 min, while for 170 °C it goes up to 52.6%. This does not mean that lignin was produced, but rather that lignin resisted more pre-hydrolysis than the remaining macromolecular compounds. Therefore, the pre-hydrolysis step increases the lignin content on the material, making it better for the production of adhesives or high-resistance foams. For α-cellulose, the values obtained at both temperatures are comparable, ranging between 30.0% and 35.7% and 31.0% and 35.6% at 160 °C and 170 °C, respectively. The reduction in holocellulose content relative to the initial sample was higher for longer times and higher temperatures. Since the differences in cellulose are not significant, the main decrease was observed in the hemicelluloses content, which decreased from 29.6% to 20.7% and 14.9% for 160 °C and 170 °C, respectively, corresponding to a percentual removal rate of 30% and 50% of the initial content.

### 3.2. FTIR Analysis of Pre-Hydrolysis

FTIR-ATR analyses were conducted on HS original material and material after pre-hydrolysis at 160 °C and 170 °C for 15 and 180 min (Figure 5). In accordance with the chemical composition, there is a reduction in carbohydrates, leading to a percentual increase in lignin. Therefore, an increase in lignin-assigned peaks and a decrease in the main carbohydrate peaks was to be expected.

Significant spectral changes were observed upon comparison of the initial material with the material after pre-hydrolysis. In the region around 3350 cm^−1^ corresponding to the OH band, there is a slight decrease and narrowing of the band, especially for 160 °C and 180 min of pre-hydrolysis. Changes in the peaks at 2930 cm^−1^ and 2850 cm^−1^ that correspond to overlapping of -CH_2_- (2935–2915 cm^−1^) and -CH_3_ (2970–2950 cm^−1^) stretching asymmetric vibrations, and -CH_2_- (2865–2845 cm^−1^) and -CH_3_ (2880–2860 cm^−1^) stretching symmetric vibrations [40], were observed, but were not very consistent. There is a slight increase for 160 °C and 170 °C (180 min) pre-hydrolysis, and there is an increase in the 2850 cm^−1^ peak in relation to 2930 cm^−1^. This increase can be due to the attack on cellulose that was observed before in the wet chemistry analysis for 180 min or due to the increased lignin percentage. A similar change was reported before for heat-treated wood, and was stated to be due to the lower frequencies of CH stretching vibrations of the methoxyl group in lignin [41,42].

In the peaks at 1730 cm^−1^ and 1600 cm^−1^, attributed to non-conjugated and conjugated C=O bonds, there seems to be a merging of the two peaks, probably due to the increase in the 1600 cm^−1^ peak, where generally lignin exhibits a strong absorption around 1600 cm^−1^ due to benzene ring stretching vibrations. This reinforces the results obtained before, showing an increase in lignin content.

The most significant changes are observed, however, in the fingerprint region with a significant increase in the 1510 cm^−1^ peak, more visible in the samples with 180 min hydrolysis. This peak is associated with benzene ring stretching vibrations [43]. Moreover, the peak is at about 1505 cm^−1^ for hardwood lignin (Guaiacyl g and Syringyl-S) and at about 1510 cm^−1^ for softwood lignin (Guaiacyl-G) [44], which suggests that HS lignin is similar to a G-lignin. Results presented before by Pérez-Armada et al. [14] suggest that HS has a GH lignin with guaiacyl (G) and Hydroxyphenil (H) moieties, contrary, for example, to walnut and almond shells, which have a GS lignin, and pine nut shells, which have a G lignin [45].

At 170 °C and 180 min, an increase of 1260 cm^−1^ was observed. This increase can be due to the increase in lignin, but this band has several different contributions. Additionally, a decrease in the 1030 cm^−1^ band, attributed to C-O stretching vibrations of C-O-C in carbohydrates, was observed.

### 3.3. Liquefaction

After the pre-hydrolysis, the remaining solid material was liquefied by polyalcohols with acid catalysis. Figure 6 presents the liquefaction yield for the original material (without pre-hydrolysis) and of the solid residues obtained after the pre-hydrolysis at 160 °C and 170 °C. Generally, the liquefaction yield of HS pre-hydrolyzed at 160 °C decreased with the time of the pre-hydrolysis, starting at around 63% for the original material and ending at 31% for the 180 min pre-hydrolyzed samples. This is probably due to the lower amount of soluble compounds in the samples, since most of the hemicelluloses were already removed. Both cellulose and lignin are less susceptible to liquefaction due to their chemical structure. The higher lignin content, denser and harder structure, and complex chemical bonding in hazelnut shells make their liquefaction more difficult. The less susceptibility for thermal degradation of shells lignin compared to other lignocellulosic materials lignin has already been reported for Ginkgo shells [46] and for palm kernel shells [47]. Similarly, for 170 °C, there is a reduction in the liquefaction yield for the pre-hydrolyzed materials during 15–60 min, but there is a posterior increase for the 120 min and 180 min pre-hydrolyzed samples, which can be due to some depolymerization of cellulose during the pre-hydrolysis step.

HS liquefaction yield is slightly lower compared to the liquefaction yield of other shells at the same conditions. For instance, the liquefaction yield of walnut shells at 180 °C and 60 min and similar polyalcohol lead to a liquefaction yield of around 80% [23]. Nevertheless, the liquefaction yield is higher than that obtained before for a liquefaction with glycerol catalyzed with sodium carbonate that needed a temperature of around 270 °C to obtain a similar liquefaction yield [48]. The liquefaction is also higher than that obtained before using water and other supercritical solvents that obtained similar percentages only at 300 °C [49].

Figure 7 presents the klason lignin of the solid residues after liquefaction using polyalcohols with acid catalysis.

The results indicate that the solid residues after liquefaction are mainly composed of lignin, with values ranging between 60–80%. However, generally, lignin and hemicelluloses are considered to be the most susceptible materials to liquefaction [50]. The high content of klason lignin can be due to the highly condensed structure that might not allow for the determination of lignin using this method. The higher amount of lignin in the solid residues of samples pre-hydrolyzed at 170 °C seems, however, to be correct, since these materials had higher lignin content after pre-hydrolysis.

### 3.4. FTIR-ATR Analysis after Liquefaction

FTIR analysis was performed in the liquefied material (LM) and in the solid residue after liquefaction (SR) (Figure 8). The temperatures (160 °C and 170 °C) and times (15 min, 180 min) refer to the pre-hydrolysis step. All the liquefied materials have a higher and broader OH stretching peak at around 3400 cm^−1^. This peak results not only from the liquefied material, but also from the polyalcohols (glycerol and ethylene glycol) used in the liquefaction procedure. Glycerol (three OH) and ethylene glycol (two OH) used in the liquefaction process significantly affect the FTIR spectra by broadening and intensifying the OH stretching peak due to extensive hydrogen bonding. This peak is not much different between the original material and the liquefaction solid residues. Both peaks at 2930 cm^−1^ and 2850 cm^−1^ stretching asymmetric and symmetric vibrations of CH, as mentioned before, are significantly different from the original HS. These peaks in LM and SR are wider and higher, with their maximum shifting for 2980 cm^−1^ and 2900 cm^−1^. The main difference is that in the liquefied material, the highest peak is at 2900 cm^−1^, and for the liquefied residue, it is 2980 cm^−1^. The bands at around 1730 cm^−1^, corresponding to non-conjugated C=O linkages, and 1600 cm^−1^, corresponding to conjugated C=O linkages, present relevant differences between the original, LM, and SR materials. The liquefied material with pre-hydrolysis at 160 °C, initially (15 min) presents a high absorption band at 1600 cm^−1^ in relation to the 1730 cm^−1^ peak. This fact and the high absorption at 1510 cm^−1^ can be due to lignin liquefaction at an early stage. The spectrum for 160 °C (180 min) has, however, a higher absorption at 1730 cm^−1^ in relation to 1600 cm^−1^. No significant changes are seen for the solid residue with 15 min or 180 min pre-hydrolysis, and similarly to the solid residue of the original material without pre-hydrolysis. There are two peaks that increase significantly in the liquefied material, which are around 1235 cm^−1^ and 1370 cm^−1^. The band around 1235 cm^−1^ can have several contributions, from skeletal vibrations in lignin, but the band around 1370 cm^−1^ has been associated with C-H bending vibrations in CH_3_ groups of acyl fragments [51].

In the spectrum of the solid residues, there are weak absorptions at 1600 cm^−1^, 1460 cm^−1^, and 1320 cm^−1^, which seem to indicate that the residues do not have high amounts of lignin as the wet chemistry, which can be due to the unsuitability of the klason method for lignin determination in these residues, as discussed before. Similar results were presented before by Silva et al. [52], who studied the liquefaction of Kraft lignin using polyhydric alcohols and organic acids as catalysts for sustainable polyols production.

The spectrum of the liquefied original material is very different from the liquefied materials with pre-hydrolysis, presenting higher peaks of OH (3350 cm^−1^), CHs (2930 cm^−1^ and 2859 cm^−1^), and C-O-C (1030 cm^−1^), mostly assigned to carbohydrates. This reinforces the results discussed before that show that pre-hydrolyzed materials have a high amount of lignin content and, therefore, a low amount of carbohydrates. The band around 1235 cm^−1^ is lower than for liquefied pre-hydrolyzed materials, which can be due to the lower percentage of phenolic compounds in the liquefied original material in comparison with the pre-hydrolyzed materials.

## 4. Conclusions

Hazelnut shell is a valuable waste material with a significant amount of lignin of around 37% (39% without extractives) and also hemicelluloses (30%). Pre-hydrolysis has been shown to be efficient in the removal of hemicelluloses without significantly solubilizing cellulose and lignin. The optimal solubilization of HS was achieved at 170 °C for 180 min, resulting in 25.8% solubilization, obtaining a final material with a 53% lignin content. This solubilized material can be later converted to small sugars, for example, using enzymatic hydrolysis.

Chemical analysis showed that the amount of hemicellulose solubilization increased with the increase in temperature and duration of the pre-hydrolysis, and that the resulting solid material had somewhat higher cellulose content and significantly higher lignin, reaching a maximum of around 53%. The higher amount of lignin seemed to have some impact on the liquefaction yield of the pre-hydrolyzed materials. Significant spectral changes in the hazelnut shells were observed using FTIR-ATR spectroscopy from their initial state to the solid residue and onto the liquefied phase, confirming the effectiveness of pre-hydrolysis in altering the material’s chemical composition and increasing the amount of lignin-derived compounds in the final liquefied material. This modification enhances the shells’ suitability for producing adhesives or polyurethane foams with higher mechanical strength. These results allowed us to obtain a polyol with improved properties while recovering some sugars at the same time. Results are underway to study the effects of the pre-hydrolysis on the physical and mechanical properties of rigid polyurethane foams from liquefied hazelnut shells.

## Figures and Tables

**Figure 1 materials-17-02667-f001:**
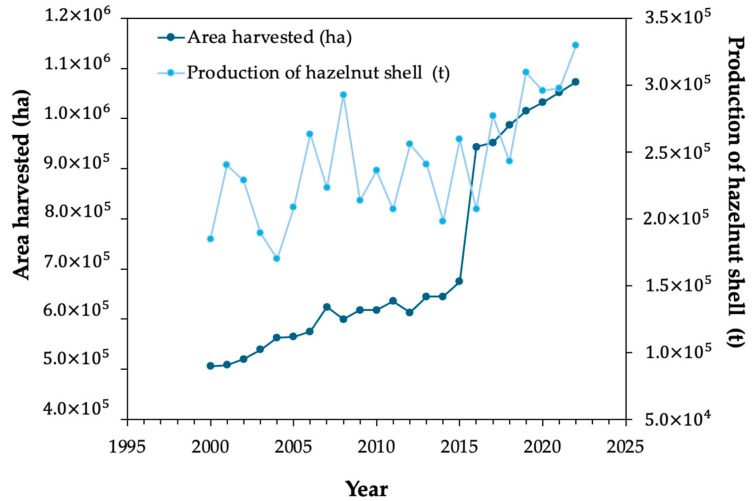
Evolution of hazelnut shell area and production worldwide [6].

**Figure 2 materials-17-02667-f002:**
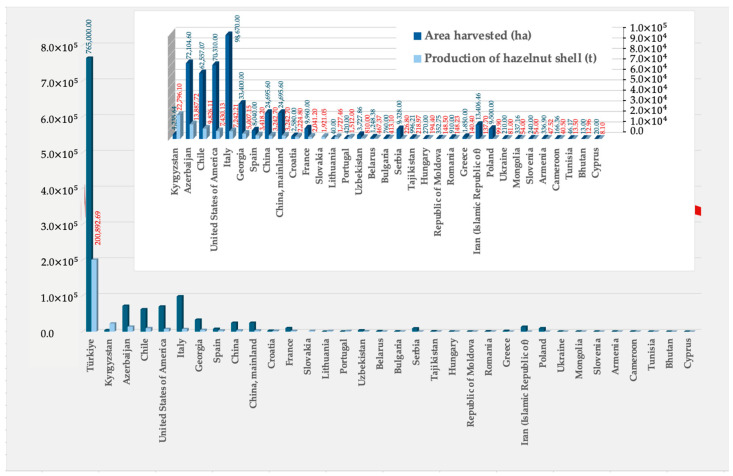
Main world hazelnut shell production in 2023 [6].

**Figure 3 materials-17-02667-f003:**
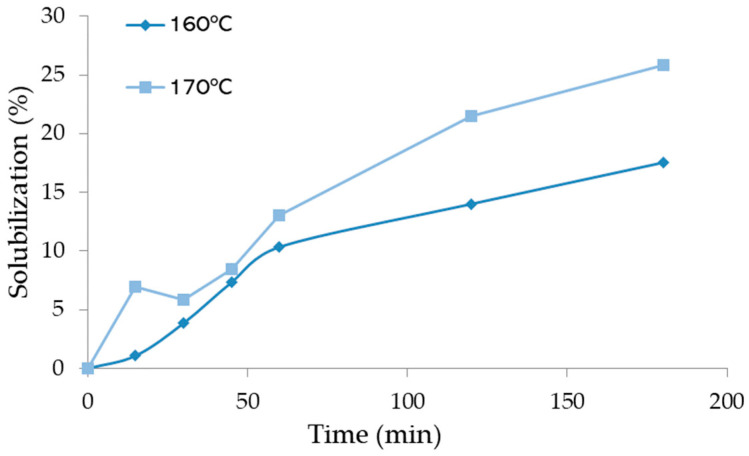
Variation of the solubilization with time in the pre-hydrolysis of HS.

**Figure 4 materials-17-02667-f004:**
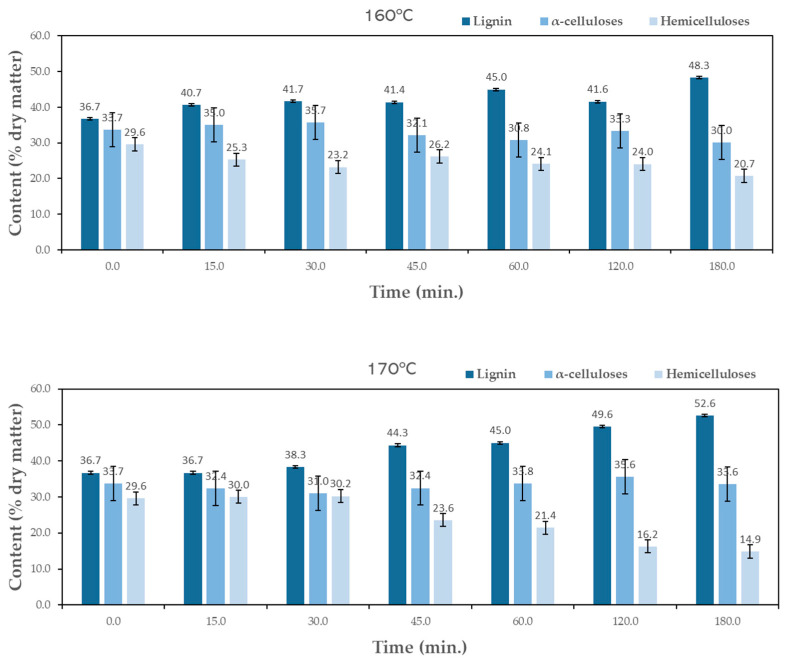
Variation in the chemical composition of the hydrolyzed HS with time at 160 °C (**top**) and 170 °C (**bottom**). The standard deviation based on five replicates is presented as error bars.

**Figure 5 materials-17-02667-f005:**
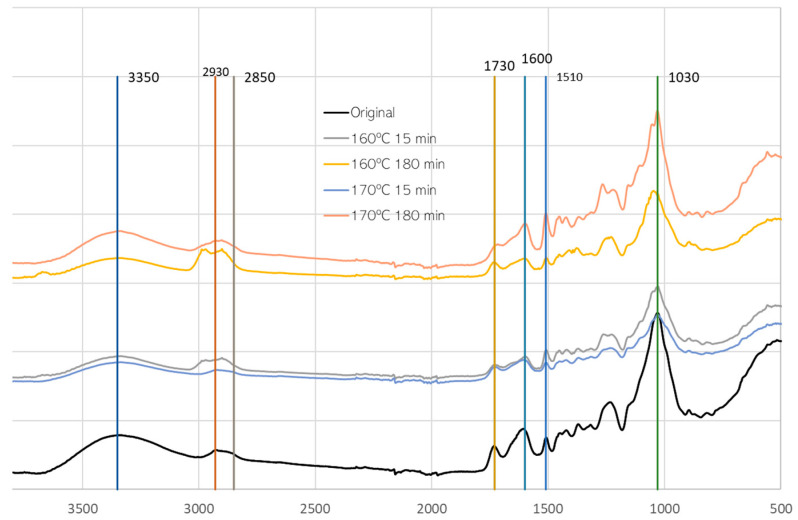
FTIR-ATR of original material and material after pre-hydrolysis at 160 °C and 170 °C.

**Figure 6 materials-17-02667-f006:**
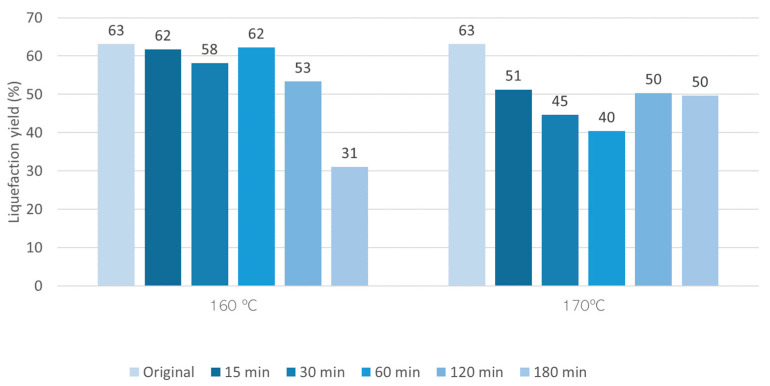
Variation of the liquefaction yield with time in the polyalcohol liquefaction of HS with and without pre-hydrolysis.

**Figure 7 materials-17-02667-f007:**
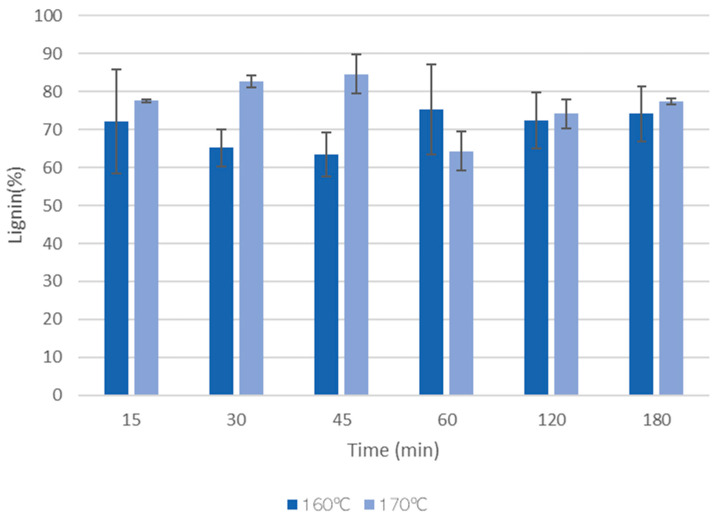
Variation of the klason lignin of the solid residues after liquefaction of HS. The standard deviation based on thee replicates is presented as error bars.

**Figure 8 materials-17-02667-f008:**
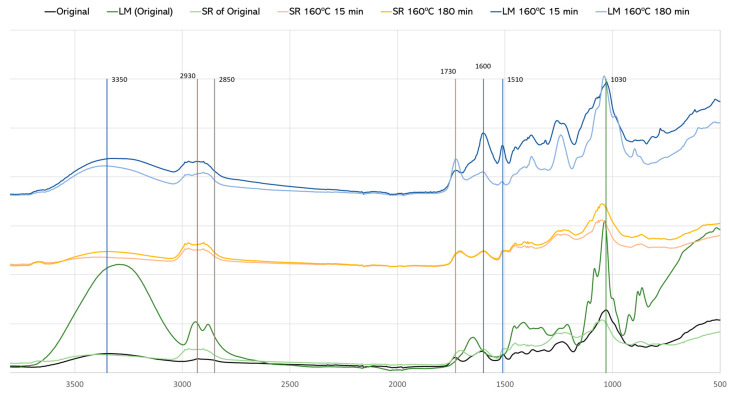
FTIR-ATR of original material and material after liquefaction.

**Table 1 materials-17-02667-t001:** Chemical composition of HS.

Parameters		Content (% Dry Matter, *w*/*w*)	Sdv
Ashes		3.65	0.15
	K	27.68	0.84
	Ca	16.86	0.48
	Mg	1.40	0.03
	Na	0.30	0.01
Extractives	Dichlorometane	0.27	0.04
	Ethanol	0.37	0.03
	Hot water	2.97	0.11
Lignin	Soluble	0.86	0.08
	Insoluble (Klason)	36.70	1.08
α-Cellulose		33.71	0.79
Hemicelluloses		29.59	2.90

## Data Availability

The original contributions presented in the study are included in the article, further inquiries can be directed to the corresponding author.

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
