# Peer review of "Enhancing Liquefaction Efficiency: Exploring the Impact of Pre-Hydrolysis on Hazelnut Shell (Corylus avellana L.)"

_materials, 2024, doi:10.3390/ma17112667_

Round 1

Reviewer 1 Report

Comments and Suggestions for Authors

Dear Authors,

I am writing to provide my review of the manuscript titled " Enhancing Liquefaction Efficiency: Exploring the Impact of Pre-Hydrolysis on Hazelnut Shell (Corylus avellana L.)" which has been submitted for publication in Materials Journal. 

This study aims to maximize Hazelnut shells 's chemical potential by solubilizing hemicelluloses to recover sugars and increase lignin content for adhesive or foam production. Pre-hydrolysis at 160 and 170°C for 15 to 180 minutes was explored. Remaining solid materials were liquefied using polyalcohols with acid catalysis. Chemical composition changes were analyzed before and after pre-hydrolysis. The process was monitored using FTIR-ATR spectroscopy. Optimal solubilization (25.8%) occurred at 170°C for 180 minutes. Hemicellulose solubilization increased with temperature and time, as did α-cellulose and lignin percentages. FTIR-ATR revealed significant spectral shifts post-hydrolysis, indicating improved material suitability for adhesives or high-strength foams.

Overall, this study contributes significantly to the development of sustainable solutions for utilizing agricultural waste, paving the way for potential applications in adhesive or foam production with enhanced mechanical properties.

During the review, I highlighted the following critical points:

Introduction section

  1. The introduction should highlight the study's specific aim of improving liquefaction efficiency through pre-hydrolysis of hazelnut shells. While the current introduction offers broad background information on hazelnuts, it fails to establish a clear link to the study's goals.

  1. Additionally, the introduction delves into excessive details about hazelnut cultivation, production, and global distribution, potentially detracting from the study's main focus. Simplifying this information could enhance the introduction's clarity and focus.

Materials and methods section

1.     The section lacks clear organization and structure, making it challenging for readers to follow the experimental procedures step by step. Reorganizing the content into distinct subsections for each experimental procedure (e.g., Materials Preparation, Pre-Hydrolysis, Liquefaction, Chemical Composition Analysis, FTIR Analysis) would enhance clarity and readability.

2.     Although the equipment used is mentioned, specific details about the experimental setup, such as the exact conditions of the reactors (e.g., pressure, heating rate), are lacking. Providing this information is crucial for reproducibility and understanding the experimental conditions.

Results and discussion section

  1. The section presents extensive data without clear organization or interpretation, making it difficult for readers to understand the key findings. Streamlining the presentation of results and focusing on the most significant observations would improve clarity and readability.

  1. While comparing the chemical composition and results of pre-hydrolysis and liquefaction with previous studies is valuable, the discussion lacks a critical analysis of the discrepancies or similarities. Providing insights into the reasons behind the differences in results between studies would enhance the discussion's depth and scientific rigor.

  1. I believe that the section could benefit from better integration of results from chemical composition analysis, pre-hydrolysis, liquefaction, and FTIR analysis. Connecting these findings and discussing their implications in a cohesive manner would strengthen the overall argument and contribute to a more comprehensive understanding of the research outcomes.

Conclusions section

  1. The conclusions provide a concise summary of the research findings, but there's a need for clearer articulation of the key results and their significance. Specifically, the impact of pre-hydrolysis on hazelnut shell composition and liquefaction efficiency should be emphasized more explicitly.

  1. It would be beneficial to include a brief section on potential future research directions or applications based on the current findings. This could help contextualize the significance of the study within the broader field and guide further investigations in this area.
Comments on the Quality of English Language

Moderate editing of English language required

Author Response

Reviewer 1

First of all, the authors would like to thank for the constructive review that will certainly improve the paper.

Introduction section

The introduction should highlight the study's specific aim of improving liquefaction efficiency through pre-hydrolysis of hazelnut shells. While the current introduction offers broad background information on hazelnuts, it fails to establish a clear link to the study's goals.

Additionally, the introduction delves into excessive details about hazelnut cultivation, production, and global distribution, potentially detracting from the study's main focus. Simplifying this information could enhance the introduction's clarity and focus.

 R: The introduction section has been reformulated. The hazelnut information has been reduced and the study goals were highlighted.

Materials and methods section

  1. The section lacks clear organization and structure, making it challenging for readers to follow the experimental procedures step by step. Reorganizing the content into distinct subsections for each experimental procedure (e.g., Materials Preparation, Pre-Hydrolysis, Liquefaction, Chemical Composition Analysis, FTIR Analysis) would enhance clarity and readability.

 R: Sorry but we did not understand what was suggested. There are already 5 sub-sections with exactly the same titles as suggested.

  1. Although the equipment used is mentioned, specific details about the experimental setup, such as the exact conditions of the reactors (e.g., pressure, heating rate), are lacking. Providing this information is crucial for reproducibility and understanding the experimental conditions.

R: Additional information was added as suggested.

Results and discussion section

The section presents extensive data without clear organization or interpretation, making it difficult for readers to understand the key findings. Streamlining the presentation of results and focusing on the most significant observations would improve clarity and readability.

 While comparing the chemical composition and results of pre-hydrolysis and liquefaction with previous studies is valuable, the discussion lacks a critical analysis of the discrepancies or similarities. Providing insights into the reasons behind the differences in results between studies would enhance the discussion's depth and scientific rigor.

 I believe that the section could benefit from better integration of results from chemical composition analysis, pre-hydrolysis, liquefaction, and FTIR analysis. Connecting these findings and discussing their implications in a cohesive manner would strengthen the overall argument and contribute to a more comprehensive understanding of the research outcomes.

 R: The discussion section was modified according to the suggested and a better connection between the different findings was attempted we hope successfully.

Conclusions section

The conclusions provide a concise summary of the research findings, but there's a need for clearer articulation of the key results and their significance. Specifically, the impact of pre-hydrolysis on hazelnut shell composition and liquefaction efficiency should be emphasized more explicitly.

 R: The conclusion section was corrected according to the suggestions.

It would be beneficial to include a brief section on potential future research directions or applications based on the current findings. This could help contextualize the significance of the study within the broader field and guide further investigations in this area.

R: Done as suggested.

Reviewer 2 Report

Comments and Suggestions for Authors

The study optimized a pre-hydrolysis process to enhance the chemical potential of hazelnut shells by solubilizing hemicelluloses and increasing lignin content, thereby making the material suitable for producing adhesives and high-strength foams. Before considering it for publication, the authors should address the following points:

Abstract

The outcome of this study is interesting, and I wonder if there are other applications for this pre-hydrolyzed material besides producing adhesives and high-strength foams.

Keywords

Isn't it better to be consistent with "hazelnut shells" instead of "bark"?

Introduction

Lines 67 to 71: The values must be rounded up; it does not make sense to use decimals with thousands of tons.

Figure 2: Review the quality of the figure. There is a vertical line on the right perimeter and a horizontal line on the top that should be removed. Additionally, the data source from 2003 is too old and should be updated to a more recent reference.

The first part of the introduction seems too long, and line 86 seems very disjointed from the subsequent sentences.

The authors should make the entire introduction more concise. There is no indication of their proposed work in the final part, referencing the structure of the paper. Please add some information in this section highlighting the novelty, goal, and scope of the study.

Materials and Methods

Equations: "initial" instead of "inicial."

Line 138: The degree Celsius symbol is not superscript; please correct this.

Results and Discussion

Table 1: The ash content decimal should be a point, not a comma. Make the line thickness consistent across each row.

Figures 4, 5, 6, 7, and 8: Please remove the perimeter lines around the images.

Could you elaborate on the extent to which the polyalcohols (glycerol and ethylene glycol) used in the liquefaction procedure influenced the observed FTIR spectra, especially the OH stretching peak at around 3400 cm-1?

Comments on the Quality of English Language

The sentences in the second part of the introduction are a bit disjointed and should be improved for better coherence and flow.

Author Response

First of all, the authors would like to thank for the constructive review that will certainly improve the paper.

The study optimized a pre-hydrolysis process to enhance the chemical potential of hazelnut shells by solubilizing hemicelluloses and increasing lignin content, thereby making the material suitable for producing adhesives and high-strength foams. Before considering it for publication, the authors should address the following points:

Abstract

The outcome of this study is interesting, and I wonder if there are other applications for this pre-hydrolyzed material besides producing adhesives and high-strength foams.

R: Yes, there are several other uses. The phrase has been rephrased. Lignin is increasingly used in the production of bioplastics and composite materials to enhance the mechanical properties and biodegradability of the resulting materials.

Keywords

Isn't it better to be consistent with "hazelnut shells" instead of "bark"?

R:This was a mistake. There is no Hazelnut bark. Corrected.

 Introduction

Lines 67 to 71: The values must be rounded up; it does not make sense to use decimals with thousands of tons.

R:Done as suggested

Figure 2: Review the quality of the figure. There is a vertical line on the right perimeter and a horizontal line on the top that should be removed. Additionally, the data source from 2003 is too old and should be updated to a more recent reference.

R:The lines were removed. The data is from 2023 as can be seen in the figure, the caption was incorrect and was corrected.

The first part of the introduction seems too long, and line 86 seems very disjointed from the subsequent sentences.

R: The introduction has been completely reformulated.

The authors should make the entire introduction more concise. There is no indication of their proposed work in the final part, referencing the structure of the paper. Please add some information in this section highlighting the novelty, goal, and scope of the study.

R: The introduction was reformulated decreasing the information about hazelnut shells and including some more information on liquefaction. The novelty, goal, and scope of the study were added as suggested

Materials and Methods

Equations: "initial" instead of "inicial."

R:Corrected

Line 138: The degree Celsius symbol is not superscript; please correct this.

R:Corrected

Results and Discussion

Table 1: The ash content decimal should be a point, not a comma. Make the line thickness consistent across each row.

R: Done according to the suggested.

Figures 4, 5, 6, 7, and 8: Please remove the perimeter lines around the images.

R: Done according to the suggested.

Could you elaborate on the extent to which the polyalcohols (glycerol and ethylene glycol) used in the liquefaction procedure influenced the observed FTIR spectra, especially the OH stretching peak at around 3400 cm-1?

R: In summary, glycerol (three OH) and ethylene glycol (Two OH) used in the liquefaction process significantly affect the FTIR spectra by broadening and intensifying the OH stretching peak around 3400 cm-1 due to extensive hydrogen bonding. Their presence also influences other spectral regions associated with C-O and C-H vibrations, although to a lesser extent.

Comments on the Quality of English Language

The sentences in the second part of the introduction are a bit disjointed and should be improved for better coherence and flow.

R: The introduction has been reformulated and we tried to increase the coherence and flow of the writing.

Reviewer 3 Report

Comments and Suggestions for Authors

This manuscript requires improvement before it can be considered for publication.

Improve English grammar and organization. (i.e spelling, word choice, sentence structure, etc)

Make sure novel aspects are detailed  throughout the text

--avoid abbreviations before they are defined...even in the abstract

--avoid lumping references - discuss each individually'

--add more current (less than 5 yrs old) references currently there are 22 of 52 references that are current

--conclusions must be discussed in more detail...please support those conclusions

--provide a detailed error and uncertainty analysis

--provide error and uncertainty details for equipment used

----is the error in fig 4 and others error or uncertainty

--use the proper number of significant digits

Comments on the Quality of English Language

Improve English grammar and organization. (i.e spelling, word choice, sentence structure, etc)

Author Response

First of all, the authors would like to thank for the constructive review that will certainly improve the paper.

This manuscript requires improvement before it can be considered for publication.

Improve English grammar and organization. (i.e spelling, word choice, sentence structure, etc).

R: The English grammar and organization has been reviewed

Make sure novel aspects are detailed throughout the text

R: The novel aspects were better detailed throughout the text trying to make the findings more connected.

--avoid abbreviations before they are defined...even in the abstract

R: The undefined FTIR-ATR abbreviation was presented as suggested.

--avoid lumping references - discuss each individually'

R:Although we were not sure to which references the reviewer meant we tried to increase the discussion on the presented references.

--add more current (less than 5 yrs old) references currently there are 22 of 52 references that are current

R: More current references were added in the introduction and on the discussion sections.

--conclusions must be discussed in more detail...please support those conclusions

R: The conclusions section has been improved.

--provide a detailed error and uncertainty analysis

R: The standard deviation was included in table 1. It was already presented in figure 4 and 7 and this was clarified in the text.

--provide error and uncertainty details for equipment used

  1. We are sorry but we did not understand. Which equipment?

----is the error in fig 4 and others error or uncertainty

R: It is uncertainty. It corresponds to the standard deviation of the results. This was clarified in the text.

--use the proper number of significant digits

R: corrected as suggested.

Round 2

Reviewer 3 Report

Comments and Suggestions for Authors

This is an improved manuscript

   There are still a few minor issues ( groups of reference should be eliminated and several minor English  grammar issues still exist) 

Comments on the Quality of English Language

   There are still a few minor issues ( groups of reference should be eliminated and several minor English  grammar issues still exist) 

Author Response

Dear Reviewer,

All the multiple citations have been removed.